# Enhancing Visual Localization with Cross-Domain Image Generation

Yuanze Wang [1]  Yichao Yan [1]  Shiming Song [2]  Songchang Jin [2]  Yilan Huang [2]  Xingdong Sheng [3]  Dianxi Shi [1 4]

## Abstract

Visual localization aims to predict the absolute camera pose for a single query image. However, predominant methods focus on single-camera images and scenes with limited appearance variations, limiting their applicability to cross-domain scenes commonly encountered in real-world applications. Furthermore, the long-tail distribution of cross-domain datasets poses additional challenges for visual localization. In this work, we propose a novel cross-domain data generation method to enhance visual localization methods. To achieve this, we first construct a cross-domain 3DGS to accurately model photometric variations and mitigate the interference of dynamic objects in large-scale scenes. We introduce a text-guided image editing model to enhance data diversity for addressing the long-tail distribution problem and design an effective fine-tuning strategy for it. Then, we develop an anchor-based method to generate high-quality datasets for visual localization. Finally, we introduce positional attention to address data ambiguities in cross-camera images. Extensive experiments show that our method achieves state-of-the-art accuracy, outperforming existing cross-domain visual localization methods by an average of 59% across all domains. Project page: https://yzwang-sjtu.github.io/CDG-Loc.

## 1. Introduction

Visual localization is the process of estimating the position and orientation of a query image in 3D space. It is crucial in applications such as robotics, autonomous driving, and

[1]MoE Key Lab of Artificial Intelligence, AI Institute, Shanghai Jiao Tong University, Shanghai, China [2]Intelligent Game and Decision Lab (IGDL), Beijing, China [3]Lenovo Research, Shanghai, China [4]Department of Big Data Intelligence, Advanced Institute of Big Data, Beijing, 100195, China. Correspondence to: Dianxi Shi <dxshi@nudt.edu.cn>.

*Proceedings of the 42nd International Conference on Machine Learning*, Vancouver, Canada. PMLR 267, 2025. Copyright 2025 by the author(s).

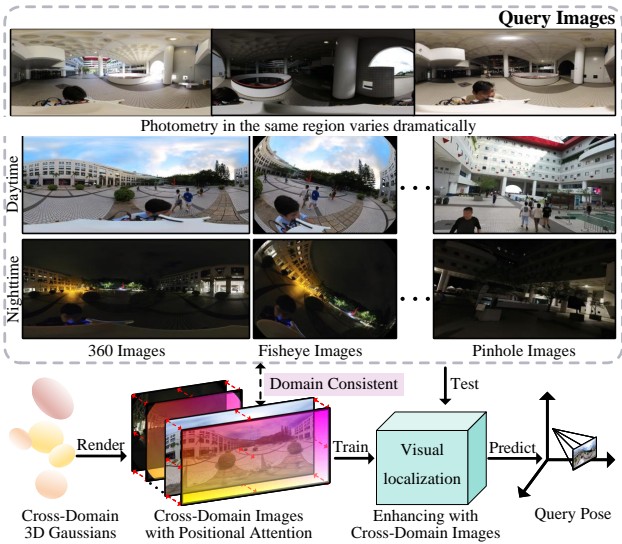

*Figure 1.* Demonstration of the proposed method. Our method achieves state-of-the-art cross-domain localization performance by rendering domain-consistent images with positional attention for data augmentation.

augmented reality. Successful visual localization methods have been developed, such as feature matching-based techniques (Zhou et al., 2022; Sarlin et al., 2021; Zhou et al., 2025), scene coordinate regression methods (Wang et al., 2023; Brachmann et al., 2023; 2025; Wang et al., 2024a), and absolute pose regression methods (Chen et al., 2024b; Shavit & Keller, 2022; Chen et al., 2022a; Shavit et al., 2021). Current methods focus on single-domain visual localization, typically applied to images from a single camera (e.g., pinhole cameras) and scenes with limited appearance variations. However, single-domain methods encounter two major challenges in real-world applications: 1) Query images captured by pinhole cameras, 360 cameras, fisheye cameras, and other cameras exhibit significant differences due to variations in distortion and field of view. 2) The appearance of scenes undergoes substantial variations over time, influenced by factors such as lighting conditions and dynamic objects. Due to the limited training data of single-domain methods, significant domain differences arise between these methods and the real-world domain, leading to a drop in localization performance. Recently, 360Loc(Huang

et al., 2024) pioneered the exploration of these challenges by mapping limited 360 images to cross-camera images for data augmentation, achieving state-of-the-art performance. However, there remains considerable room for improvement due to the limited diversity of the generated data.

In light of these challenges, our research focuses on developing an advanced cross-domain dataset generation method to enhance visual localization performance, as illustrated in Fig. 1. This paper concentrates on data augmentation for absolute pose regression (APR) methods, which require image-pose pairs as training datasets. To achieve this, we adopt 3D Gaussian Splatting (3DGS) (Kerbl et al., 2023) for its capability to synthesize high-fidelity images from arbitrary views in real time, thus enhancing the diversity of data for visual localization. However, generating cross-domain datasets based on 3DGS faces two main challenges: 1) **Photometric variations and dynamic objects** disrupt the multi-view consistency required by 3DGS methods, making it difficult to reconstruct accurate and domain-consistent scene appearances. 2) Due to the high costs in time and labor, training datasets often exhibit a **long-tail distribution**, which prevents 3DGS from fully learning the scene representations across all domains. For instance, data collected under daytime conditions is abundant (**primary domain**), while data from nighttime conditions is extremely sparse (**secondary domain**) in our evaluated dataset.

To address these challenges, we propose a novel cross-domain 3D Gaussian Splatting to model large-scale scene photometric variations while mitigating the interference of dynamic objects. We introduce learnable photometric embeddings that encode image photometric histograms to represent scene photometry. These embeddings are integrated with the anchor-based Gaussian features proposed by Scaffold-GS(Lu et al., 2024) to model scene photometric variations, achieving strong scene fitting and reduced Gaussian storage. Additionally, we introduce dynamic photometric embeddings, Gaussian dynamic confidence, and Gaussian dynamic photometry to ignore dynamic objects and compensate for photometric losses. Unlike WildGaussians (Kulhanek et al., 2024), our method encodes photometric histograms and is not constrained by the performance bottlenecks of pre-trained detectors (Oquab et al., 2023). Considering that sparse secondary domain images hinder 3DGS from accurately learning cross-domain scene representations, we employ a text-guided image editing model (Brooks et al., 2023) to augment secondary domain images by transforming primary domain images. Recognizing the domain inconsistencies between the pre-trained editing model and real-world domains, as well as the presence of hallucination noise, we carefully design an efficient fine-tuning strategy to adapt the editing model to our task. Furthermore, we propose a two-phase training strategy for cross-domain 3DGS to mitigate hallucination noise introduced by the editing model.

To ensure that cross-domain 3DGS generates high-quality images for visual localization, we develop an anchor-based image generation method. This method guarantees that the generated images are correctly associated with photometric embeddings and uniformly distributed within the range of accurate reconstruction. We then generate cross-camera images by leveraging the camera models. These images exhibit different distortions and fields of view, leading to the misalignment of appearance features across images corresponding to the same pose. This introduces **data ambiguity**, making it difficult for visual localization methods to learn effectively. To address this, we propose an efficient positional attention mechanism to align appearance features, thereby eliminating data ambiguity.

Extensive experiments demonstrate that our method achieves state-of-the-art visual localization performance, with more than 59% average improvement across domains. Our main contributions are summarized as follows:

- We developed a cross-domain 3DGS to generate real-domain consistent images by modeling photometric variations and mitigating the interference of dynamic objects in large-scale scenes. Furthermore, we designed a two-stage training strategy to reduce the interference of hallucination noise.

- We introduce a text-guided image editing model to enhance data diversity for addressing the long-tail distribution problem and design an efficient fine-tuning strategy to adapt the model to our task.

- We develop an anchor-based method to generate high-quality datasets for visual localization and introduce positional attention to eliminate data ambiguity.

## 2. Related Work

### 2.1. Visual Localization

Several studies have explored cross-domain visual localization under pinhole camera settings. To enhance localization performance, (Porav et al., 2018) uses invertible generators to produce synthetic images, while (Anoosheh et al., 2019) converts nighttime images to a more discriminative daytime representation. Other methods focus on learning domain-invariant features to bridge the gap between varying environmental conditions (Hu et al., 2019). Additionally, several works (Tang et al., 2020; Chen et al., 2023) advocate for disentangling image representations into separate codes that isolate place-specific cues from appearance and occlusion factors, ensuring reliable place recognition. However, these methods are limited to single-camera localization and rely on image retrieval. Compared to APR methods

(Clark et al., 2017; Moreau et al., 2022; Chen et al., 2024a), the retrieval-based methods suffer from significantly higher computational costs and storage requirements due to the need to construct and maintain a retrieval database. This paper enhances APR-based cross-domain localization, including cross-camera scenarios, by proposing a novel cross-domain image generation method. Unlike the prior method (Huang et al., 2024) with limited image augmentation, our method facilitates diverse cross-domain image generation.

## 2.2. Neural Rerendering for Unconstrained Scenes

Recent advances in neural rendering have enabled the 3D reconstruction from unconstrained scenes. Neural Rerendering in the Wild (Meshry et al., 2019) combines traditional 3D reconstruction with neural networks to handle unconstrained scenes. Extensions to the Neural Radiance Field (NeRF) (Martin-Brualla et al., 2021; Chen et al., 2022b) further address challenges in unconstrained scenes by embedding appearance information and transient uncertainty. In addition, methods such as Neural Scene Chronology (Lin et al., 2023) focus on capturing temporal variations by employing modules for appearance hallucination and temporal step function encoding. However, their slow training and rendering make large-scale scene modeling and data generation time-consuming. Additionally, the limited parameters of NeRF (Mildenhall et al., 2021) hinder its ability to effectively represent large outdoor scenes. Recently, 3DGS-based methods have garnered attention due to their faster optimization and rendering efficiency compared to NeRF. Among them, SpotLessSplats (Sabour et al., 2025), GS-W (Zhang et al., 2025), WE-GS (Wang et al., 2024b), and WildGaussians (Kulhanek et al., 2024) have shown potential in modeling appearance variations and dynamic objects in wild scenes. However, they cannot explicitly control photometric properties and are constrained by the performance bottlenecks of pre-trained detectors (Oquab et al., 2023). Existing methods are also not suitable for scenes with long-tail distribution problems. In this paper, we propose a method that models appearance variations by encoding photometric histograms, mitigates the impact of dynamic objects without relying on pre-trained detectors, and employs a fine-tuned image editing model to effectively address the long-tail distribution problem.

## 2.3. Image Editing

Traditional methods focused on tasks like style transfer (Gatys, 2015; Liu et al., 2024; Ojha et al., 2021), while recent approaches incorporate text-based guidance (Bao et al., 2023) (e.g., CLIP embeddings) to enhance editing. Recent progress in image editing (Zhang et al., 2023; Chefer et al., 2023; Li et al., 2024; Ceylan et al., 2023) leverages large pretrained models and generative techniques for targeted manipulations. Text-to-image diffusion models like Stable

Diffusion (Rombach et al., 2022) enable complex transformations through realistic image generation from text prompts. In 3D scene editing, instruction-guided methods (Chen et al., 2024c; Haque et al., 2023) allow manipulation of scene elements, such as object location, shape, and lighting, using natural language instructions. The state-of-the-art techniques (Tschernezki et al., 2022) distill knowledge from generated multi-view images using pre-trained models. However, the images generated directly using the pre-trained model are misaligned with the target domain, which undermines localization accuracy. To address this, we carefully design an effective fine-tuning strategy to train the editing model to generate domain-consistent images.

## 3. Preliminaries

Scaffold-GS (Lu et al., 2024) uses a set of 3D anchor structures to model the entire scene. Each anchor stores learnable Gaussian features $f_{gs}$ and employs a multilayer perceptron to predict the properties of Gaussians inherited from the anchor representation. Compared to the original 3D Gaussian structure (Kerbl et al., 2023), Scaffold-GS offers a more compact representation and stronger scene modeling capabilities, allowing for flexible modeling of dynamic scene appearances. Similar to traditional 3DGS, it utilizes tile-based rasterization for efficient image rendering. The geometry of each 3D Gaussian $G_i$ is defined in world space:

$$G_i(\mathbf{x}) = e^{-\frac{1}{2}(\mathbf{x}-\mathbf{p}_i)^T \Sigma_i^{-1}(\mathbf{x}-\mathbf{p}_i)}, \quad (1)$$

where $\mathbf{x}$ is an arbitrary position within the 3D scene, $\mathbf{p}_i$ and $\Sigma_i$ denote the position and covariance matrix of 3D Gaussians, respectively. $\Sigma_i$ uses semi-definite parameterization to constrain itself to the space of valid covariance matrices:

$$\Sigma = RSS^T R^T, \quad (2)$$

where $R$ is rotation matrix and $S$ is scaling matrix. Additionally, each 3D Gaussian includes opacity $\alpha_i$ and color $c_i$ for image rendering. To render an image from a given view, 3D Gaussians are first projected onto the image plane to obtain 2D Gaussians $G_i'$. The 2D Gaussians are then processed using tile-based rasterization and alpha-blending to produce the final rendered image:

$$C(\mathbf{x}') = \sum_{i \in N} c_i \alpha_i G_i'(\mathbf{x}') \prod_{j=1}^{i-1}(1 - \alpha_j G_j'(\mathbf{x}')), \quad (3)$$

where $\mathbf{x}'$ and $N$ denote the queried pixel position and the number of 2D Gaussians associated with the queried pixel, respectively. All parameters of the 3D Gaussians are optimized end-to-end using a differentiable rasterizer.

## 4. Method

We aim to develop a visual localization method to accurately estimate the poses of cross-domain query images, given a

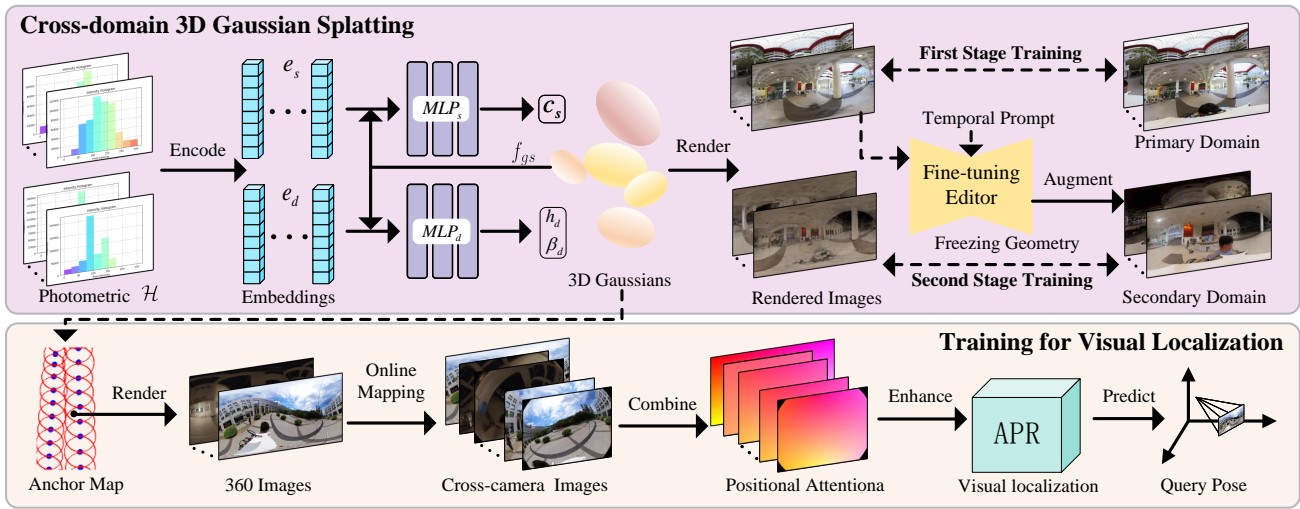

*Figure 2.* **Overview of our method.** We first construct a cross-domain 3DGS to accurately model photometric variations and mitigate the interference of dynamic objects. Specifically, we construct learnable embeddings to encode photometric histograms. These embeddings, along with Gaussian features, are fed into two separate MLPs to predict the static and dynamic attributes of the Gaussians. We fine-tune an image editing model to generate secondary domain images for augmenting the sparse secondary domain dataset. Furthermore, we propose a two-stage training strategy for cross-domain 3DGS to mitigate the hallucination noise introduced by the editing model. Next, we develop an anchor-based generation method to create high-quality datasets for visual localization. Finally, we implement an online mapping mechanism to reduce data storage costs and introduce positional attention to resolve data ambiguities.

limited training set consisting of 360 image-pose pairs as shown in Fig. 2. To achieve this, we first construct a cross-domain 3DGS to accurately model photometric variations and mitigate the interference of dynamic objects in large-scale scenes. Next, we introduce image editing models that transform images from the primary domain to augment sparse secondary domains, addressing the long-tail distribution problem. Finally, we propose an anchor-based method to generate high-quality datasets for visual localization and introduce positional attention to resolve data ambiguities.

### 4.1. Cross-Domain 3D Gaussian Splatting

To accurately model photometric variations and ignore the interference from dynamic objects in the large scenes, we carefully design an efficient cross-domain 3DGS. The main components of this module include: (a) photometric variation modeling, (b) suppression of dynamic objects, and (c) a two-phase training strategy.

**Photometric Variation Modeling.** The photometry in images varies significantly with position and time in real-world scenes, leading to the failure of 3DGS that depends on multi-view consistency. To address this challenge, we construct $L$ learnable photometric embeddings $e$ uniformly distributed across the entire scene based on mapping images. These embeddings encode the photometric histogram $\mathcal{H}$ of the corresponding mapping images, introducing photometric

priors that vary with position and time. We employ the multi-layer perceptron (MLP) to predict Gaussian attributes by combining Gaussian features $f_{gs}$ proposed by Scaffold-GS (Lu et al., 2024) with the photometric embeddings, thereby achieving strong scene modeling capabilities. Please refer to the supplementary materials for more details.

**Dynamic Object Suppression.** Dynamic objects introduce noise that hinders the accurate reconstruction of scenes. To suppress the influence of dynamic objects, we introduce dynamic confidence $\beta$ for the Gaussian attributes and render dynamic confidence maps $M$ using alpha-blending:

$$M(x) = \sum_{i \in N} \beta_i \sigma_i \prod_{j=1}^{i-1} (1 - \sigma_j), \qquad (4)$$

Inspired by NeRF-W (Martin-Brualla et al., 2021), we assign lower weights to the training loss in the regions with high dynamic confidence to ignore the dynamic objects. However, varying photometry is partially treated as dynamic objects and thus ignored, distorting the photometric rendering of the scene. To address this, we introduce a dynamic photometric embedding $e_d$ to incorporate photometric variation priors, assisting in the prediction of dynamic confidence. We also add a dynamic photometric $h_d$ to the Gaussians to compensate for photometric loss. To prevent entanglement, two separate MLP networks $MLP_s$, $MLP_d$ are used to

predict static and dynamic attributes independently:

$$c_s = MLP_s(e_s, f_{gs}),$$
$$h_d, \beta = MLP_d(e_d, f_{gs}),$$ (5)

where $c_s$ and $e_s$ represent static color and static photometric embeddings. We can obtain the rendered image $\hat{\mathbf{C}}$:

$$\hat{\mathbf{C}}(x) = \sum_{i \in N}(c_s + h_d)\sigma_i \prod_{j=1}^{i-1}(1 - \sigma_j),$$ (6)

**Two-phase Training Strategy.** Since the training dataset consists of only 360 images, we first convert them into pinhole camera images for training the cross-domain 3DGS, then map the rendered pinhole images back to 360 images. Due to the significant photometric differences in cross-domain images and hallucination noise in the augmented secondary domain images (Sec. 4.2), directly training cross-domain 3DGS on all domain images makes it difficult to learn accurate scene representations. Therefore, we train cross-domain 3DGS in two stages. In the first stage, we only use images from the primary domain, learning the accurate appearance of the primary domain and the global scene geometry. Inspired by NeRF-W (Martin-Brualla et al., 2021), we use D-SSIM loss and the photometric loss with uncertainty to mitigate the interference of dynamic objects:

$$\mathcal{L} = \lambda(\frac{\|\mathbf{C} - \hat{\mathbf{C}}\|_2^2}{2M^2} + \frac{\log M^2}{2}) + (1 - \lambda)\mathcal{L}_{\text{D-SSIM}},$$ (7)

Where $\lambda$ is the loss weight, $\mathbf{C}$ and $\hat{\mathbf{C}}$ represent the ground truth and the rendered result, respectively. In the second stage, we fine-tune the first-stage 3DGS using augmented secondary domain images. Specifically, we freeze parameters related to scene geometry and only train parameters related to photometric variations. This prevents hallucination noise from damaging the already reconstructed scene geometry. Finally, we use $\mathcal{L}_1(\mathbf{C}, \hat{\mathbf{C}})$ and LPIPS losses (Haque et al., 2023) effectively reduce the hallucination noise.

## 4.2. Secondary Domain Augmentation.

The scarcity of secondary domain data due to the long-tailed distribution problem of cross-domain datasets limits the ability of cross-domain 3DGS to learn accurate scene representations. We introduce a text-guided image editing model to address the long-tail distribution problem and design an efficient fine-tuning strategy to adapt the model to our task, as shown in Fig. 3. Inspired by the powerful InstructPix2Pix (Brooks et al., 2023) based on diffusion, we define a temporal prompt $T_c$ for temporal transformation (e.g., *make it nighttime*) and leverage the editing model to transform primary domain images into the secondary domain. However, directly using the pre-trained model results in domain inconsistencies. To address this, we fine-tune

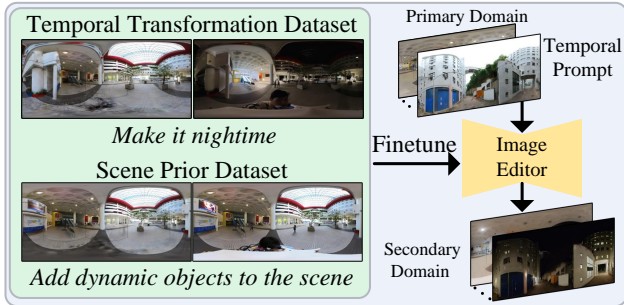

*Figure 3.* Illustration of secondary domain augmentation.

the image editing model using a carefully constructed **temporal transformation dataset**. Specifically, we render the primary domain images $I_a$ corresponding to the sparse secondary domain images $I_b$ using the first-stage 3DGS. Then, the temporal transformation datasets pair the original images $I_a$, the edited images $I_b$, and the editing text $T_c$. We further randomly rotate the 360 images $I_o$ by the latitude $\theta_i$ and longitude $\phi_i$ using the 360-camera projection function $\psi_o$ for dataset augmentation:

$$\hat{I}_o = \psi_o(R(\theta_i, \phi_i)\psi_o^{-1}(I_o)).$$ (8)

Considering that sparse temporal transformation datasets fail to cover the entire global scene, leading to hallucination phenomena when the model encounters unseen areas. We constructed a **scene prior dataset**, which also helps prevent dynamic objects in the fine-tuning dataset from introducing dynamic noise into the model. Specifically, the scene prior datasets pair the rendered static images and dynamic images from the primary domain with editing text, such as *add dynamic objects to the scene*.

### 4.3. Training for Visual Localization

**Anchor-based Image Generation Method.** To generate high-quality datasets for training visual localization methods, we need to ensure that the generated images are correctly associated with photometric embeddings and uniformly distributed within the range of accurate reconstruction. Therefore, we develop an anchor-based image generation method. Specifically, photometric embeddings uniformly distributed across the scene are used as anchors, with each anchor associated with a pose $P_a$, where $P_a$ denotes the pose of the image encoded by the photometric embedding. Then, we randomly generate $m$ poses $P_g$ within a 3D spherical range with a radius $L$, centered at the anchor. The radius is empirically determined as the distance between adjacent anchors to ensure that the generated images fall within the range of accurate reconstruction. The pose $P_g$ is computed using random displacements $t$ and random rotation $R$:

$$P_g = RP_a + t, \quad \|t - t_a\| \leq L,$$ (9)

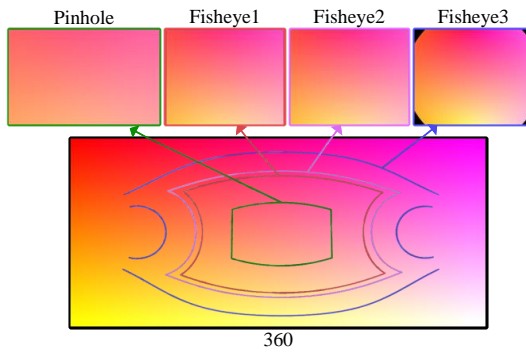

*Figure 4.* Visualization of positional attention mapping across different camera domains in the global coordinate space.

where $t_a$ represents the position of the anchor. Finally, we render the image using the pose $P_g$ and the photometric embedding of its anchor. This ensures that the correct photometric embedding is associated during image rendering, which is crucial for generating domain-consistent images.

**Online Cross-Camera Images Generation.** The generated images are currently in the 360 format, we need to generate cross-camera images. Thanks to the full field of view of 360 images, any camera image can be directly mapped from the 360 images using the mapping $\Phi_i$:

$$I_{c_n} = \Phi_i(I_o) = \psi_{c_n}(\psi_o^{-1}(I_o)), \quad (10)$$

where $\psi_o^{-1}$ denote the unprojection function of 360 camera and $\psi_{c_n}$ is the projection function of camera domain $c_n$. If there are $j$ camera domains, this requires generating $m \times j$ images for each anchor. However, storing images for all camera domains incurs high storage costs. To address this, we store only the mapping relationships $\Phi_i$, rendering the cross-camera images online during the training of the visual localization method. Since the mapping relationships are precomputed offline and dominate the computational cost of cross-camera mapping, the additional overhead introduced by online mapping is acceptable.

**Positional Attention.** The appearance features of cross-camera images with the same pose are not aligned in the image coordinate space due to different distortions and fields of view. We define this problem as data ambiguity, which lowers the accuracy of visual localization methods. To address this, we propose position attention to guide the alignment of appearance features. First, we construct a global image coordinate space $S^o$ based on 360 images, which can cover the field of view of any camera domain. Then, we convert the image coordinates $S_{c_n}$ from different camera domains to the global coordinate space $S_{c_n}^o$ as shown in Fig. 4, which are used as position attention:

$$S_{c_n}^o = \psi_o(\psi_{c_n}^{-1}(S_{c_n})). \quad (11)$$

*Table 1.* The settings for the mapping and query datasets.

| | **Mapping** | | | |
|---|---|---|---|---|
| Scene | Atrium | Concourse | Hall | Piatrium |
| Primary/Secondary (Frames) | 581/10 | 491/10 | 540/20 | 632/20 |
| Camera Model | 360 | 360 | 360 | 360 |
| Spatial Extent (m$^2$) | 2340 | 1395 | 5460 | 6860 |
| | **Query** | | | |
| Camera Model | 360 | Fisheye1/Fisheye2/Fisheye3 | | Pinhole |
| Field of View | 360° | 120°/150°/195° | | 85° |
| Resolution | 6144×3072 | 1280×1024 | | 1920×1200 |

We obtain the normalized positional attention $S_{c_n}^n$ by using the dimensions $D$ of the global coordinate space:

$$S_{c_n}^n = \frac{2S_{c_n}^o}{D-1} - 1. \quad (12)$$

Finally, the positional attention maps are concatenated with the images along the channel and fed into the visual localization method, guiding cross-camera image alignment.

## 5. Experiments

### 5.1. Datasets and Baselines

We conduct experiments using the large-scale dataset 360Loc (Huang et al., 2024), which includes dynamic objects and significant variations in lighting conditions. The dataset consists of five camera domains, covering both daytime (primary domain) and nighttime (secondary domain). We construct a long-tailed distribution dataset by selectively sparse secondary domain data based on scene size. The settings for the mapping and query datasets are detailed in the Tab. 1. To facilitate training and evaluation, we use SAM2 (Ravi et al., 2024) to remove the fixed base at the bottom of the image. For comparison, we employ state-of-the-art baselines provided by cross-domain visual localization benchmarks (Huang et al., 2024). To ensure a fair comparison, we reproduce the baselines using the same setup as our method. Please refer to the supplementary materials for more details.

### 5.2. Implementation Details

Our method is built upon the widely-used open-source Scaffold-GS (Lu et al., 2024) codebase and employs the MS-T (Shavit et al., 2021) for visual localization. During cross-domain 3DGS training, we scale images to 400×400 and train for 60,000 iterations in the first stage, followed by fine-tuning for 20,000 iterations in the second stage. For the editing model, we fine-tune it for 10,000 iterations at a resolution of 512×512. The number $L$ of photometric embeddings is set to half the number of the mapping dataset and the parameter $m$ is set to 24. During visual localization training,

*Table 2.* Comparison of median translation and orientation error ($m$, $^\circ$) on the primary domain dataset.

| Scene | Atrium_Day | | | | | | Concourse_Day | | | | | |
|---|---|---|---|---|---|---|---|---|---|---|---|---|
| Camera | Pinhole | Fisheye1 | Fisheye2 | Fisheye3 | 360 | Average | Pinhole | Fisheye1 | Fisheye2 | Fisheye3 | 360 | Average |
| PN (Kendall et al., 2015) | 15.5/85.5 | 11.8/79.9 | 11.3/80.4 | 10.1/79.8 | 4.5/82.6 | 10.6/81.6 | 18.6/95.0 | 10.7/88.7 | 10.1/87.7 | 7.7/85.9 | 2.6/39.0 | 9.9/79.3 |
| MS-T (Shavit et al., 2021) | 20.1/77.7 | 14.5/73.7 | 14.0/71.7 | 12.5/65.6 | 13.2/54.7 | 14.9/68.7 | 17.0/69.1 | 11/60.8 | 10.5/59.6 | 9.9/63.4 | 4.9/34.6 | 10.7/57.5 |
| PN-VC2 (Huang et al., 2024) | 11.5/31.2 | 9.1/19.5 | 8.9/19.3 | 8.4/18.1 | 7.7/17.1 | 9.1/21.0 | 7.0/18.9 | 4.7/11.5 | 4.4/11.2 | 3.9/9.9 | 3.2/6.9 | 4.6/11.7 |
| MS-T-VC2 (Huang et al., 2024) | 10.0/46.7 | 6.3/36.3 | 6.2/36.5 | 4.9/28.1 | 5.2/47.6 | 6.5/39.0 | 3.5/19.6 | 1.9/12.7 | 1.8/12.8 | 1.7/12.7 | 1.6/13.5 | 2.1/14.3 |
| **Ours** | **5.2/21.8** | **2.2/12.7** | **2.1/12.3** | **1.8/11.5** | **1.6/14.1** | **2.6/14.5** | **2.7/9.7** | **1.5/5.7** | **1.3/5.5** | **1.1/5.4** | **1.0/5.7** | **1.5/6.4** |
| Scene | Hall_Day | | | | | | Piatrium_Day | | | | | |
| Camera | Pinhole | Fisheye1 | Fisheye2 | Fisheye3 | 360 | Average | Pinhole | Fisheye1 | Fisheye2 | Fisheye3 | 360 | Average |
| PN (Kendall et al., 2015) | 19.8/90.8 | 11.1/95.7 | 10.4/95.9 | 7.5/92.3 | 2.7/97.3 | 10.1/96.1 | 21.3/88.0 | 13.8/86.5 | 13.0/87.1 | 10.9/84.4 | 5.7/69.8 | 12.9/83.2 |
| MS-T (Shavit et al., 2021) | 21.8/96.0 | 14.1/94.0 | 13.5/92.7 | 11.2/91.7 | 11.4/88.5 | 14.4/92.6 | 30.0/78.5 | 23.4/71.3 | 22.7/72.7 | 19.3/72.6 | 11.3/52.0 | 21.3/69.4 |
| PN-VC2 (Huang et al., 2024) | 7.0/26.4 | 4.4/15.5 | 4.3/14.7 | 3.9/14.2 | 3.2/12.2 | 4.6/16.6 | 10.4/24.5 | 6.9/13.3 | 6.8/12.8 | 6.8/11.5 | 5.0/9.8 | 7.2/14.4 |
| MS-T-VC2 (Huang et al., 2024) | 4.0/36.9 | 2.2/25.8 | 2.1/25.2 | 1.8/26.2 | 1.7/26.9 | 2.4/28.2 | 8.1/32.7 | 4.0/24.5 | 3.9/23.6 | 3.8/24.1 | 3.3/22.1 | 4.6/25.4 |
| **Ours** | **2.3/17.4** | **1.0/8.9** | **1.0/8.6** | **0.9/8.4** | **0.7/7.5** | **1.2/10.2** | **4.4/18.0** | **2.5/11.3** | **2.4/10.5** | **2.4/10.1** | **2.3/10.1** | **2.8/12.0** |

*Table 3.* Comparison of median translation and orientation error ($m$, $^\circ$) on the secondary domain datasets.

| Scene | Atrium_Night | | | | | | Concourse_Night | | | | | |
|---|---|---|---|---|---|---|---|---|---|---|---|---|
| Camera | Pinhole | Fisheye1 | Fisheye2 | Fisheye3 | 360 | Average | Pinhole | Fisheye1 | Fisheye2 | Fisheye3 | 360 | Average |
| PN (Kendall et al., 2015) | 19.8/90.8 | 16.4/87.3 | 16.1/86.7 | 15.7/83.5 | 8.7/22.8 | 15.3/74.2 | 19.8/96.2 | 13.3/85.5 | 11.1/87.5 | 11.1/87.5 | 5.5 / 45.6 | 12.5/80.6 |
| MS-T (Shavit et al., 2021) | 24.8/84.7 | 17.8/80.5 | 17.0/80.1 | 15.1/80.0 | 5.0/46.3 | 15.9/74.3 | 22.1/74.7 | 17.7/69.5 | 17.0/68.3 | 17.2/72.7 | 8.4/57.3 | 16.5/68.5 |
| PN-VC2 (Huang et al., 2024) | 13.6/50.2 | 11.5/38.1 | 11.6/38.3 | 11.4/ 37.8 | 9.7/45.7 | 11.6/42.0 | 10.7/25.4 | 8.3/ 16.4 | 8.4/15.7 | 8.1/ 14.7 | 5.5/13.5 | 8.2/17.1 |
| MS-T-VC2 (Huang et al., 2024) | 13.5/70.6 | 9.1/64.1 | 9.0/63.8 | 8.6/ 64.4 | 7.3/ 61.5 | 9.5/64.9 | 5.5/28.7 | 4.0/ 20.1 | 3.9/20.3 | 3.5/20.9 | 3.4/23.7 | 4.1/22.7 |
| **Ours** | **4.6/25.3** | **1.8/11.0** | **1.7/10.1** | **1.6/10.1** | **1.4/7.3** | **2.2/12.8** | **3.3/11.4** | **1.7/6.6** | **1.6/6.3** | **1.4/6.9** | **1.1/6.1** | **1.8/7.5** |
| Scene | Hall_Night | | | | | | Piatrium_Night | | | | | |
| Camera | Pinhole | Fisheye1 | Fisheye2 | Fisheye3 | 360 | Average | Pinhole | Fisheye1 | Fisheye2 | Fisheye3 | 360 | Average |
| PN (Kendall et al., 2015) | 22.8/95.6 | 18.3/93.8 | 17.9/93.6 | 16.8/91.3 | 10.9 /100.1 | 17.3/94.9 | 33.2/89.7 | 29.3/89.6 | 28.8/90.4 | 29.2/89.8 | 30.6/98.5 | 30.2/91.6 |
| MS-T (Shavit et al., 2021) | 32.1/96.1 | 29.9/94.2 | 29.7/93.6 | 31.8/93.1 | 22.3/94.4 | 29.2/94.3 | 37.7/91.3 | 37.3/89.1 | 36.2/89.0 | 35.7/85.6 | 37.2/83.0 | 36.8/87.6 |
| PN-VC2 (Huang et al., 2024) | 13.1/56.1 | 10.5/43.7 | 10.1/43.0 | 9.7/40.5 | 7.5/34.7 | 10.2/43.6 | 15.8/51.9 | 13.2/ 34.9 | 12.8/ 34.8 | 12.5/ 32.6 | 10.6/ 30.9 | 13.0/37.0 |
| MS-T-VC2 (Huang et al., 2024) | 8.9/73.1 | 5.6/ 67.6 | 5.3/ 66.3 | 4.8/ 68.9 | 3.8/ 67.2 | 5.7/68.6 | 16.0/ 63.9 | 13.0/ 53.8 | 12.4/53.7 | 11.9/,49.9 | 11.9/,49.9 | 13.0/54.2 |
| **Ours** | **4.2/32.5** | **2.2/18.5** | **2.1/17.5** | **1.9/15.7** | **1.5/18.0** | **2.4/20.4** | **7.2/27.9** | **4.6/16.1** | **4.3/16.1** | **3.7/14.1** | **2.6/9.6** | **4.5/16.8** |

images from all domains are resized to 256×256. Training and evaluation are performed on an NVIDIA GeForce GTX 4090 GPU. Please refer to the supplementary materials for more details.

### 5.3. Quantitative and Qualitative Results

**Quantitative Results.** To evaluate the performance of cross-domain localization, we apply the widely-used median translation (m) and rotation ($\circ$). The cross-domain localization performance is reported in Tab. 2 for daytime and Tab. 3 for nighttime. Our method achieves state-of-the-art results across all cross-domain scenes, significantly outperforming the previously leading MS-T-VC2 by an average of 51% during the day and 67% at night. Additionally, our method significantly outperforms the original MS-T without data augmentation, achieving an 85% improvement in domain average. This demonstrates that visual localization is highly data-hungry, particularly in more challenging large-scale cross-domain scenes. Although MS-T-VC2 also utilizes data augmentation methods, it relies on limited mapping data, which restricts the diversity of the augmented data. Furthermore, the baselines fail to address the long-tail problem of cross-domain data, preventing visual localization methods from learning sufficient scene features and leading to poor performance in secondary domains. This emphasizes the importance of augmenting data diversity

for visual localization and validates the effectiveness of our secondary domain augmentation method.

According to the cross-camera localization results, we observe that visual localization performance decreases as the field of view (FOV) of the camera narrows. Notably, for pinhole images with a small FOV, localization performance is significantly worse than the average of the other four cameras with a wide FOV (5.2/21.8 vs. 1.9/12.7 in Atrium_Day), even when effective data augmentation methods are employed. The poor performance of pinhole images significantly lowers the overall average localization accuracy of our method. This demonstrates that visual localization performance faces a bottleneck due to the inherent limitations of a narrow FOV. These findings suggest that selecting cameras with a wide FOV is crucial for achieving better performance in practical visual localization applications.

**Qualitative Results.** The Fig. 5 shows the cross-domain image generation results of our method, covering both diverse appearance domains and camera domains. It is evident that our method generates domain-consistent images across all target domains, even in highly sparse secondary domain scenes. Moreover, the generated images successfully ignore dynamic interference to ensure accurate scene reconstruction, which is essential for achieving robust visual localization performance. These results demonstrate that

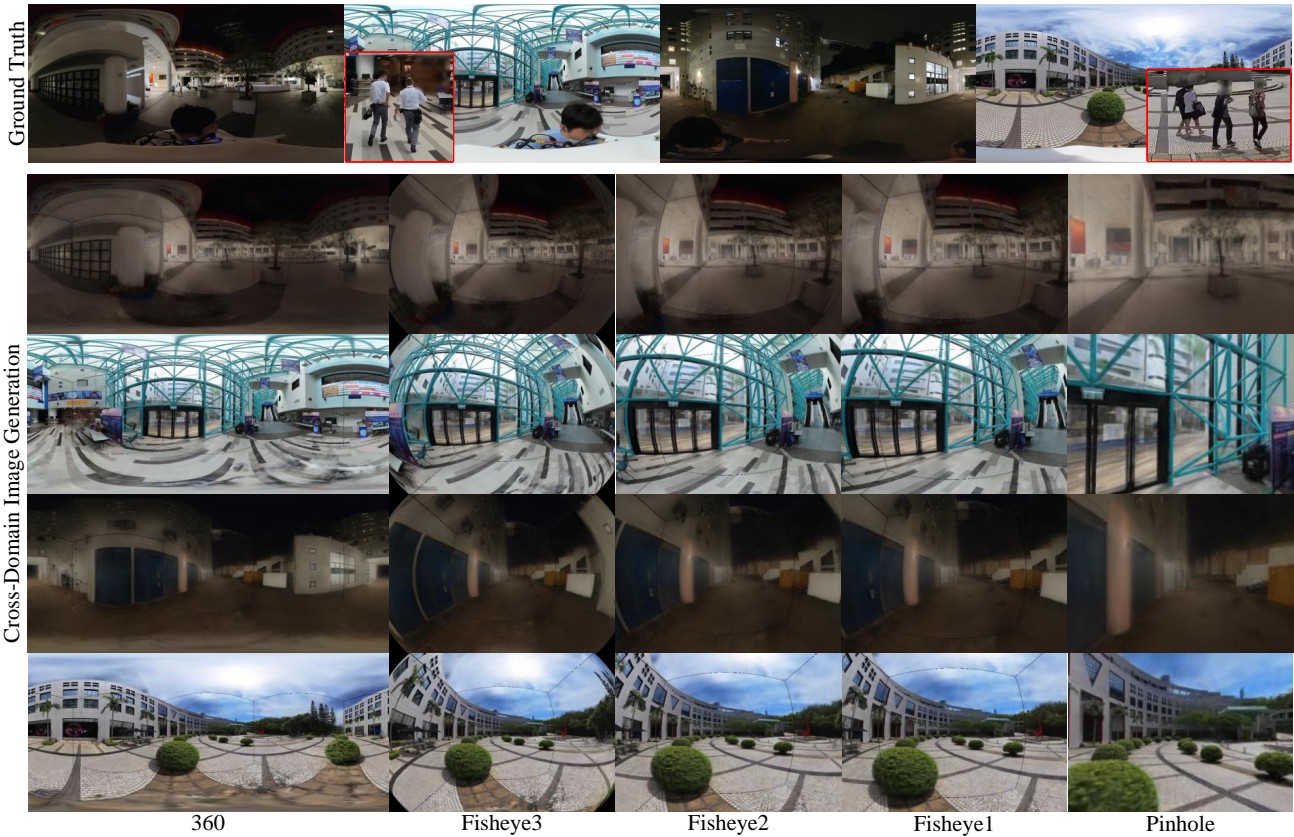

*Figure 5.* Cross-domain image generation results. Our method can generate accurate and domain-consistent images, even in sparse secondary domain scenes. Furthermore, our generated images effectively ignore dynamic interference to ensure accurate scene reconstruction.

the proposed method can generate high-quality and diverse training data for cross-domain visual localization.

### 5.4. Ablation Study

**Effectiveness of Cross-Domain 3DGS.** To evaluate the effectiveness of cross-domain 3DGS, we conducted a qualitative comparison with the Scaffold-GS in terms of dynamic interference removal, photometric variation modeling, and sparse secondary domain scene reconstruction, as shown in Fig. 6. Our method achieves accurate static appearance reconstruction by effectively suppressing dynamic interference while precisely reconstructing domain-consistent appearances in regions with significant photometric variations. As shown in the bottom row of the figure, the baseline fails to model scenes in the sparse secondary domain, while our method successfully reconstructs domain-consistent scenes. We perform a quantitative comparison using PSNR and LPIPS metrics on the Atrium datasets. As shown in Tab. 4, our method significantly outperforms the baseline. This demonstrates that cross-domain 3DGS can accurately reconstruct scene representations across all domains, even in the sparse secondary domain.

*Table 4.* Ablation study on cross-domain 3DGS with metrics "PSNR ↑ / LPIPS ↓". The symbol "/" indicates an invalid result.

| Domain | day | night | Domain | day | night |
| --- | --- | --- | --- | --- | --- |
| Scaffold-GS | 22.38/0.38 | / | **Ours** | **24.64/0.23** | **26.07/0.22** |

**Effectiveness of Fine-Tuning Strategy.** We effectively fine-tune the image editing model to generate domain-consistent images while reducing hallucinations and dynamic noise. To better demonstrate the effectiveness of this strategy, we perform a qualitative comparison with a pre-trained model, as shown in Fig. 7. The top row shows that the pre-trained editing model fails to generate domain-consistent images, such as in photometric distribution and color. In contrast, our fine-tuned model generates domain-consistent images. Furthermore, as illustrated in the middle and bottom rows, our fine-tuned model effectively prevents the introduction of hallucinations and dynamic noise. We also conducted quantitative experiments on the atrium scene. We employed image similarity metrics SSIM and LPIPS to evaluate the

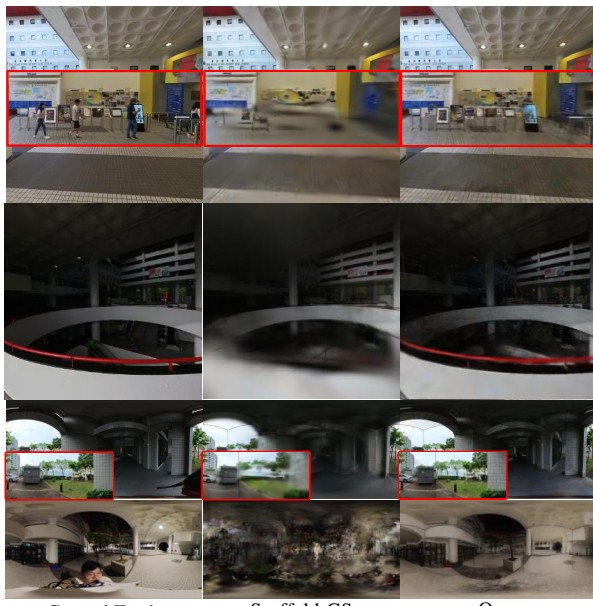

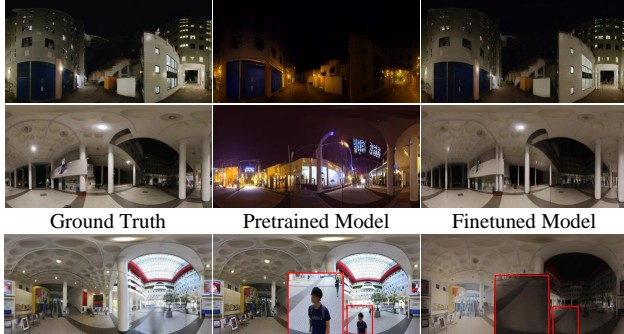

Figure 7. Ablation study on fine-tuning strategy. Top row: Our fine-tuned model generates domain-consistent images compared to the pre-trained model. Middle row: Our fine-tuned model reduces hallucination noise relative to the pre-trained model. Bottom row: Our method introduces scene priors to the model without introducing dynamic noise.

Ground Truth     Scaffold-GS     Ours

Figure 6. Ablation study on cross-domain 3DGS. The top two rows show qualitative comparisons in scenes with dynamic object interference and significant photometric variations, respectively. The bottom two rows show the comparison results for panoramic images. The red box indicates the zoomed-in region.

Table 5. Ablation study on fine-tuning strategy.

| Metrics | SSIM ↑ | LPIPS ↓ |
|---|---|---|
| Pretrained Model | 0.338 | 0.428 |
| **Finetuned Model** | **0.709** | **0.337** |

Table 6. Ablation study on positional attention on primary datasets.

| Camera | Pinhole | Fisheye1 | Fisheye2 | Fisheye3 | 360 | Average |
|---|---|---|---|---|---|---|
| w/o Positional Attention | 4.7/20.6 | 2.4/12.9 | 2.3/12.6 | 2.3/12.6 | 2.1/14.4 | 2.8/14.6 |
| **Full Model** | **3.7/16.7** | **1.8/9.7** | **1.7/9.2** | **1.6/8.9** | **1.4/9.4** | **2.0/10.8** |

similarity between the images generated and the ground truth. As shown in the Tab. 5, the image similarity significantly improved after fine-tuning. It demonstrates that our fine-tuning strategy effectively reduces the domain inconsistencies between the pre-trained editing model and real-world domains.

**Effectiveness of Positional Attention.** Positional attention effectively resolves data ambiguity by aligning appearance features of cross-camera images in global coordinate space, thereby enhancing visual localization performance. To evaluate the effectiveness of positional attention, we assess the visual localization performance without positional attention in the primary domain scenes, as shown in Tab. 6. The positional attention significantly improves overall localization performance (2.0/10.8 vs. 2.8/14.6), demonstrating that it effectively resolves data ambiguity.

## 6. Limitation

In real-world scenarios, the geometric structure of the scene may change over time. Significant geometric changes within the query images can impair cross-domain localization performance. Future improvements could involve continual learning to adapt the model to evolving scene geometry.

## 7. Conclusion

In conclusion, this work presents a novel cross-domain data generation method designed to enhance visual localization performance. We develop a cross-domain 3DGS that accurately models photometric variations while effectively suppressing dynamic object interference in large-scale scenes. We introduce an image editing model to enhance data diversity for addressing the long-tail distribution problem and design an effective fine-tuning strategy for it. Furthermore, we propose an anchor-based dataset generation method to produce high-quality training data for visual localization and incorporate positional attention to address data ambiguities in cross-camera images. Experimental results demonstrate that our method significantly improves the performance of visual localization methods, providing a promising direction for advancing cross-domain visual localization.

## Impact Statement

This paper presents work whose goal is to advance the field of Machine Learning. There are many potential societal consequences of our work, none which we feel must be specifically highlighted here.

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

# A. Implementation Details

**Cross-Domain 3D Gaussian Splatting.** We use static photometric embeddings with a dimension of $L \times 5$ and dynamic photometric embeddings with a dimension of $L \times 2$, while the photometric histogram has a dimension of 10. The photometric embeddings encoding the histogram are concatenated with the Gaussian features and fed into two separate MLPs to predict the static and dynamic attributes of the Gaussians. Each MLP has a depth of 2 and hidden layer dimensions of 64. To avoid division by zero in the training loss, we add a bias of 0.1 to the dynamic confidence map. For the first stage of cross-domain 3DGS, we set the loss weight $\lambda = 0.8$. The loss weight of the second stage is set to 1 for both components. We utilize even-indexed images for training the cross-domain 3DGS, while using odd-indexed images for evaluation purposes. We map panoramic images to pinhole images using cubemap projections to train the 3DGS.

For data augmentation, we rotate the temporal transformation dataset by 8 randomly sampled angles. During fine-tuning of the instruct-pix2pix editing model, we set the random flip, mixed precision, an initial learning rate of $\lambda = 10^{-5}$, and a batch size of 4. When converting primary domain data to secondary domain data, we set the image-guided and text-guided weights to 1.5 and 7, respectively, and configure 50 iterations for inference.

**Training for Visual Localization.** The original mapping dataset (Huang et al., 2024) does not contain any secondary domain (nighttime) scenes. Therefore, we construct a long-tailed distribution mapping dataset by selecting secondary domain images from the query set based on scene size. The specific sources of the selected images are shown in Tab. 7. To prevent data leakage, the selected mapping images are removed from the query images. For a fair comparison, we reproduce all baselines on the long-tailed distribution data constructed in this paper. Additionally, all images from all domains are resized to 256×256. We set an initial learning rate of $\lambda = 10^{-4}$ and a batch size of 32 for 300 epochs for both our method and the baselines.

**Relationship between Rendering Quality and Localization Performance.** To evaluate the impact of rendering quality, we conduct a quantitative experiment on the Atrium scene. Specifically, we compare the original Scaffold-GS (which produces lower-quality renderings compared to our cross-domain 3DGS) combined with our proposed data generation and positional attention mechanism. Localization performance drops (2.4/13.65 vs. 5.95/27.85) when using the lower-quality images generated by Scaffold-GS. This result demonstrates a positive correlation between rendering quality and localization accuracy.

*Table 7.* The specific sources of sparse secondary domain images.

| atrium/query_360/nighttime_360_1 | | | | | | | | | |
|---|---|---|---|---|---|---|---|---|---|
| 24 | 58 | 114 | 221 | 226 | 283 | 353 | 394 | 527 | 576 |
| concourse/query_360/nighttime_360_0 | | | | | | | | | |
| 2 | 53 | 132 | 155 | 232 | 257 | 321 | 366 | 410 | 461 |
| hall/query_360/nighttime_360_1 | | | | | | | | | |
| 12 | 54 | 80 | 110 | 136 | 162 | 188 | 214 | 240 | 266 |
| 292 | 318 | 348 | 374 | 403 | 426 | 452 | 555 | 582 | 607 |
| piatrium/query_360/nighttime_360_0 | | | | | | | | | |
| 2 | 26 | 50 | 74 | 98 | 122 | 146 | 170 | 194 | 218 |
| 254 | 266 | 362 | 386 | 410 | 434 | 458 | 482 | 663 | 674 |

# B. More Results

**Effectiveness of Cross-Domain 3DGS.** We have included comparative experiments with the advanced GS-W (Zhang et al., 2024), which is designed for unconstrained scenes. Since GS-W does not address the long-tail data distribution issue, we conduct quantitative experiments on the Atrium_Day scene, which contains sufficient data. Additionally, we extend its training iterations to 140,000 for the large-scale evaluation scene. As shown in the Tab. 8, our method significantly outperforms GS-W while using fewer iterations (60,000). This demonstrates that our method has stronger scene modeling capabilities for large-scale scenes.

*Table 8.* Ablation study on fine-tuning strategy.

| Metrics | PSNR ↑ | SSIM ↑ | LPIPS ↓ |
|---|---|---|---|
| GS-W (Zhang et al., 2024) | 18.85 | 0.635 | 0.594 |
| **Cross-domain 3DGS (Ours)** | **24.64** | **0.758** | **0.231** |

**Effectiveness of Anchor-Based Image Generation.** To ensure that randomly generated images for visual localization are correctly associated with photometric embeddings and uniformly distributed within the range of accurate reconstruction, we develop an anchor-based image generation method. To validate the effectiveness of anchor-based approach, we compare it qualitatively with the method that randomly associates photometric embeddings. As shown in Fig. 8, it is evident that random association of photometric embeddings often leads to inconsistency with the photometric distribution of the current pose, resulting in domain-inconsistent images. Since we first render pinhole images and then map them to 360 images, this process exacerbates the aforementioned photometric inconsistencies, leading to issues such as the formation of dark or bright spots. In contrast, our method is associated with accurate photometric embeddings, generating domain-consistent and globally photometrically consistent images. This demonstrates the effectiveness of our proposed anchor-based image generation method.

**Effectiveness of Fine-Tuning Strategy.** We designed an effective fine-tuning strategy to train the image editing model (Brooks et al., 2023) in order to transfer the pre-trained domain to the real domain. To demonstrate the effectiveness of the proposed fine-tuning strategy, we compare it qualitatively with the pre-trained model, as shown in Fig. 9. It is evident that the pre-trained model not only generates images that are inconsistent with the real domain in terms of photometric and color distribution but also suffers from hallucination effects due to the lack of scene prior knowledge. In contrast, our method generates domain-consistent images and reduces hallucination noise by incorporating scene priors.

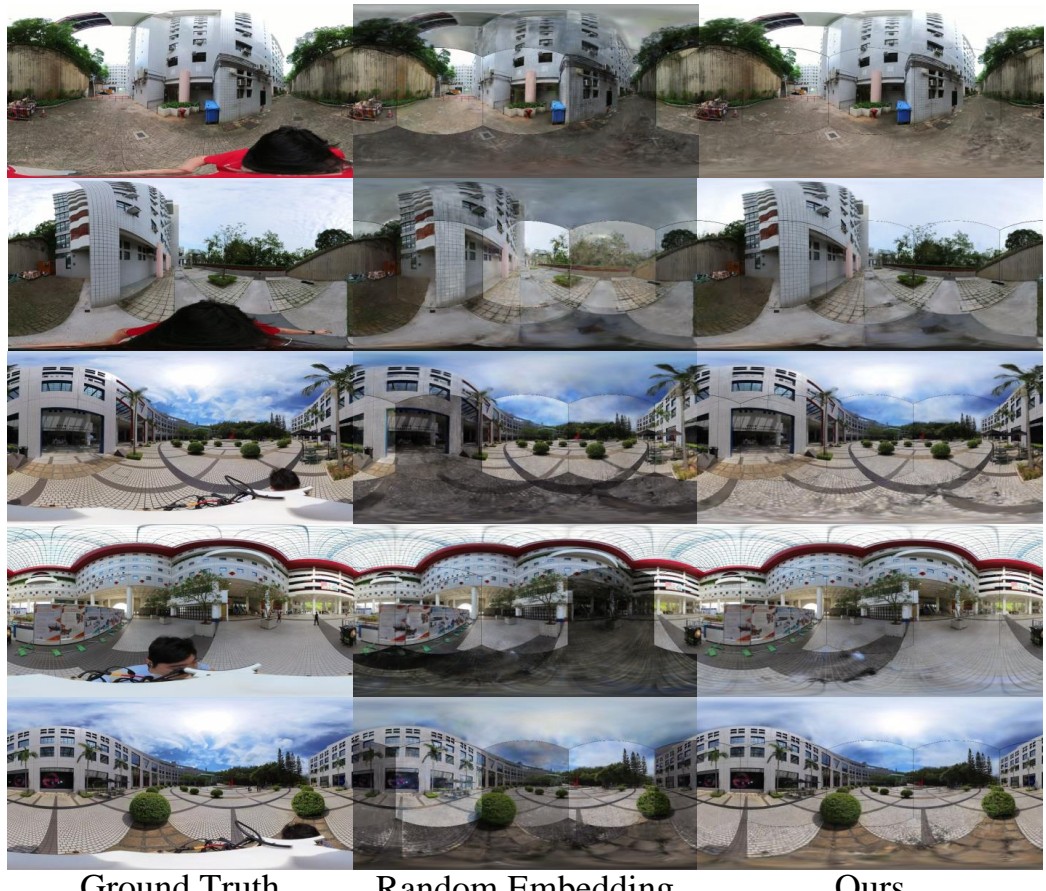

Ground Truth        Random Embedding        Ours

*Figure 8.* Ablation study on anchor-based image generation, we observe that using random embeddings to generate images results in photometric inconsistencies with the current pose, leading to global photometric artifacts such as bright and dark spots. In contrast, our method accurately associates the generated images with precise photometric embeddings, ensuring globally photometrically consistent images. The experimental results validate the effectiveness of the Anchor-Based Image Generation method.

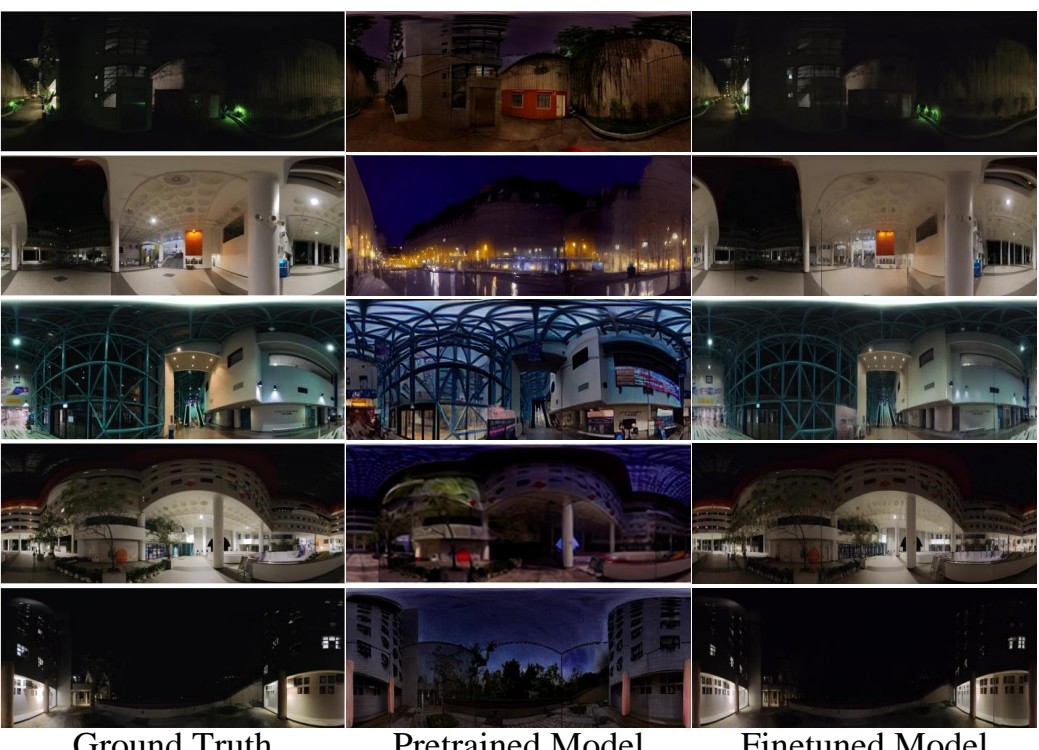

Ground Truth          Pretrained Model          Finetuned Model

*Figure 9.* Ablation study on the fine-tuning strategy, we observe that the pre-trained model not only generates images that are inconsistent with the real domain but also suffers from hallucination effects due to the lack of scene information. In contrast, our fine-tuned model generates domain-consistent images and reduces hallucination noise by incorporating scene prior knowledge.

