# OpenReview forum: "Enhancing Visual Localization with Cross-Domain Image Generation"
_ICML.cc/2025/Conference — ICML 2025 poster_

### Official Review · Reviewer_MirD · 2025-03-07

**Overall Recommendation:** 3

**Summary:**

Summary
This paper proposes a novel cross-domain data generation framework to enhance visual localization in scenarios with significant domain variations. The main results look solid and impressive. The contributions include 1) A modified 3D Gaussian Splatting framework that models photometric variations via learnable embeddings, suppresses dynamic objects using confidence maps, and employs a two-stage training strategy to mitigate hallucination noise. 2) A text-guided image editing model (fine-tuned with scene priors) to augment sparse secondary domains (e.g., nighttime data). 3) A method to synthesize pose-consistent training data and positional attention to resolve cross-camera feature misalignment.

**Claims And Evidence:**

Yes.

**Essential References Not Discussed:**

No.

**Experimental Designs Or Analyses:**

The experimental designs are sound, including comparisons on dataset 360Loc (Atrium Day, Concourse Day, Hall Day, Piatrium Day, Atrium Night, Concourse Night, Hall Night, Piatrium Night).
The ablation is also conducted on Effectiveness of Cross-Domain 3DGS, Effectiveness of Fine-Tuning Strategy, and Effectiveness of Positional Attention.

**Methods And Evaluation Criteria:**

Yes.

**Other Comments Or Suggestions:**

None.

**Other Strengths And Weaknesses:**

Strengths: 1) The practical focus on real-world challenges including cross-camera and long-tail data.

Weakness: 1) Limited discussion of failure cases.
                   2) No analysis of computational costs.
                   3) Is there any possibility to include more comparison methods?

**Questions For Authors:**

None.

**Relation To Broader Scientific Literature:**

1) 3DGS: Extends Scaffold-GS with photometric modeling by addressing limitations in dynamic scenes.
2) ​NeRF-W: Adopts uncertainty-aware rendering for dynamic object suppression.
​3) InstructPix2Pix: Leverages diffusion models for domain transfer and adds scene-specific fine-tuning.

**Theoretical Claims:**

No Theoretical Claims.

---

> ### Author Rebuttal · Authors · 2025-03-30
>
> **1 Limited discussion of failure cases.**
>
> Thank you for your valuable suggestions.
> We observe two main failure cases in our current method. First, in scenes like Piatrium_Night that contain extremely dark regions, the fine-tuned image editing model sometimes fails to produce realistic textures. This highlights the limitation of fine-tuned InstructPix2Pix in handling low-light appearances. Second, when significant static geometry changes are present in the query images (such as structural modifications), our cross-domain localization performance may degrade. In the future, this issue could be addressed by incorporating static geometry editing into the cross-domain image generation process, thereby enhancing data diversity and improving robustness in such scenarios. We have updated the manuscript to include the discussion of failure cases.
>
> **2 No analysis of computational costs.**
>
> We apologize for the missing details. All experiments were conducted on a single NVIDIA RTX 4090 GPU. Fine-tuning the image editing model for each scene takes approximately 10 hours on average. Cross-domain 3DGS training and image generation require at most 4 hours per scene. Visual localization training takes roughly 16 hours per scene. During inference, the localization method MS-T with the proposed positional attention mechanism runs at 32.51 ms per frame. We have updated the manuscript to include the computational costs of the experiments.
>
> **3 Is there any possibility to include more comparison methods?**
> 1. For the cross-domain visual localization task, our work primarily focuses on Absolute Pose Regression (APR) methods, and the baselines we compare against are the state-of-the-art APR methods provided by the 360Loc benchmark. To the best of our knowledge, there are currently no other advanced methods targeting this task.
>
> 2. For the image generation task, we have included comparative experiments with the advanced GS-W [1], which is designed for unconstrained scenes. Since GS-W does not address the long-tail data distribution issue, we conduct quantitative experiments on the Atrium_Day scene, which contains sufficient data. Additionally, we extend its training iterations to **140,000** for the large-scale evaluation scene.
>
> |                         | PSNR ↑ | SSIM ↑  | LPIPS ↓ |
> |-------------------------|--------|---------|---------|
> | GS-W                    | 18.85  | 0.635   | 0.594   |
> | **Cross-domain 3DGS（Ours）** | **24.64**  | **0.758**   | **0.231**   |
>
> As shown in the table, our method significantly outperforms GS-W while using fewer iterations (**60,000**). This demonstrates that our method has stronger scene modeling capabilities for large-scale scenes.
>
> [1] Gaussian in the Wild: 3D Gaussian Splatting for  Unconstrained Image Collections [ECCV 2024]

---

### Official Review · Reviewer_yDcV · 2025-03-11

**Overall Recommendation:** 3

**Summary:**

The paper focuses on improving visual localization accuracy with cross-domain image generation by three contributions. First,  a crossdomain is developed 3DGS to to generate realdomain consistent images. Second, a text-guided image editing model is presented to enhance data diversity for addressing the long-tail distribution problem. Third, an anchor-based method is developed to generate highquality datasets for visual localization. Extensive experiments demonstrate that the method improve visual localization performance on Loc360 dataset.

**Claims And Evidence:**

Insufficiently. The authors claim that the proposed text-guided image editing model enhances data diversity for addressing the long-tail distribution problem. In my opinion, the validation is not sufficient. To validate the effectiveness of fine-tuning strategy, the paper provides some visualization results, such as Fig. 7 and Fig. 2 in the Supplementary Material. However, the quantitative results are missing. How the strategy improve actual localization accuracy is not discussed.

**Essential References Not Discussed:**

Essential References are discussed.

**Experimental Designs Or Analyses:**

The experiment is somewhat insufficient at the following aspects.
1. Limitation results on 360 image. In Tables 1,2, the localization improvements on 360Loc are presented. It is suggested to add the results of the common localization dataset.
2. Limitation results on APR methods. The experiments results demonstrate the improvement of APR methods. Additionally, SCR methods always show more accurate performance than APR methods. Is the proposed method can enhance localization accuracy of SCR methods?
3. More quantitative results of ablation studies are preferred. To validate the effectiveness of fine-tuning strategy, the paper presents visualization results. However, the quantitative result is missing.

**Methods And Evaluation Criteria:**

Yes, the proposed methods can be used for enhancing visual localization performance.

**Other Comments Or Suggestions:**

The comments and suggestions are listed in the Strengths And Weaknesses part.

**Other Strengths And Weaknesses:**

Strength.

1.A cross-domain 3D GS to generate real-domain consistent images.

2.A text-guided image editing model to enhance data diversity.

3.An anchor-based method to generate high-quality datasets for visual localization.

Weakness.

1.The contributions seem incremental.

(1)The proposed Cross-Domain 3D Gaussian Splatting is based on Scaffold-GS with dynamic object suppression and training strategy, which is engineered.

(2)The proposed text-guided image editing model seems a application of structPix2Pix. The original contribution of the paper is not clear.

2.The writing needs further improvements.

(1)The of Introduction is somewhat confused, which is hard to read. It is suggested to rewrite it to express the challenge and contributions more clearly.

(2)In Section Method, it is suggested to present the overview of the whole method, especially where the paper contributions. Meanwhile, Fig.2 is also suggested to be modified.

3.The motivation for APR method. The pipeline of localization enhancement is designed for APR method. However, as far as current works cover, APR methods are always less accurate than SCR methods. Why not use the cross-domain image generation for SCR methods.

4.The experiments are somewhat insufficient. Please see the section “Experimental Designs Or Analyses” for details.

**Questions For Authors:**

In Tables 1,2, the paper presents the localization accuracy with cross-domain image generation. There are two questions. Which method does the paper use and How many generated images are used? Please present more details.

**Relation To Broader Scientific Literature:**

The main contribution lie in the usage of cross-Domain image generation with 3D GS and image editing model for enhancing visual localization.

**Theoretical Claims:**

N/A

---

> ### Author Rebuttal · Authors · 2025-03-30
>
> **1. Limitation results on 360Loc.**
>
> Thank you for your valuable suggestion. Our work targets the challenging task of cross-domain visual localization, which requires datasets containing query images captured by **various types of cameras**. To the best of our knowledge, 360Loc is the only existing benchmark with cross-camera query images. Therefore, we have exclusively selected 360Loc as our evaluation benchmark.
>
> **2. Why not use the cross-domain image generation for SCR methods? Is the proposed method can enhance localization accuracy of SCR methods?**
>
> Thank you for your valuable question. We would like to clarify the following points: 1) The baseline methods we evaluated follow the 360Loc benchmark, which does not include SCR methods for comparison. 2) In theory, our method could enhance the accuracy of SCR by using cross-domain 3DGS to render paired RGB and depth images. 3) Adapting SCR to cross-domain tasks is more complex, as SCR relies on **camera intrinsics and distortion parameters**, such as scene coordinate calculations, reprojection, and pose solving via PnP. However, existing SCR methods are designed for **pinhole cameras**. Adapting SCR to 360° and fisheye cameras requires deriving tailored methods. These enhancements to SCR are beyond the scope of our work, but it is a promising direction for future research.
>
> **3. More quantitative results to validate the effectiveness of the fine-tuning strategy.**
>
>  Thank you for your valuable suggestion. To validate the effectiveness of our fine-tuning strategy, we conducted quantitative experiments on the atrium scene. We employed image similarity metrics SSIM and LPIPS to evaluate the similarity between the images generated and the ground truth.
> | | SSIM ↑ | LPIPS ↓ |
> |------------|--------|---------|
> | Pretrained | 0.338  | 0.428   |
> | Finetuned  | **0.709**  | **0.337**   |
>
> As shown in the table, the image similarity significantly improved after fine-tuning, which demonstrates the effectiveness of the proposed fine-tuning strategy.
>
> **4. The proposed Cross-Domain 3DGS is engineered.**
>
> Our method builds upon Scaffold-GS but introduces key innovations to address challenges like appearance variations, dynamic object interference, and long-tail data distributions. We propose a photometric modeling scheme combining histogram-based embeddings and compensation to handle appearance changes, incorporate photometric priors for dynamic object uncertainty, and fine-tune a text-guided image editing model to augment sparse datasets in long-tail distributions. These improvements form the core innovation of our Cross-Domain 3DGS.
>
> **5. The proposed text-guided image editing model seems an application of InstructPix2Pix. The original contribution of the paper is not clear.**
>
> To address the long-tail distribution issue in the dataset, we use InstructPix2Pix to transform daytime images into sparse nighttime images for data augmentation. However, directly using the pre-trained model leads to domain inconsistencies and hallucinations, as shown in Figure 7. To overcome this, we propose a fine-tuning strategy. We use 3DGS to render daytime images corresponding to sparse nighttime images, creating a domain shift dataset that enables InstructPix2Pix to learn domain-consistent transformations. Additionally, to mitigate hallucinations due to the sparse domain shift dataset, we generate a scene prior dataset by rendering daytime images that cover the entire scene, enriching the model with more scene priors.
>
> **6. It is suggested to rewrite the Introduction to clearly express the challenge and contributions, provide an overview of the entire method, and modify Fig. 2.**
> 1. Thank you for your constructive suggestions. We have revised the Introduction to better highlight the core challenges of cross-domain visual localization and our contributions. We begin by emphasizing the challenges in cross-domain localization, followed by the motivation for introducing 3DGS to address these issues. We then discuss the technical challenges in designing cross-domain 3DGS and the solutions we developed. Finally, we discuss the issues encountered when training visual localization methods with data generated based on 3DGS and how our method is designed to overcome these training issues.
> 2. We have rewritten a more structured overview of our method and modified Figure 2 to more clearly express the design of cross-domain 3DGS and the training of the visual localization method. We have also updated the paper to include the Impact Statement.
>
> **7. In Tables 1,2, which method does the paper use, and how many generated images are used?**
> 1. Sorry for the unclear details. The method used in the paper is MS-T.
> 2. We provide the detailed counts used for each scene (including self-rotated images and ground truth).
> | Atrium | Concourse | Hall  | Piatrium |
> |--------|-----------|-------|----------|
> | 17,430  | 14,730     | 16,200 | 18,960   |
>
> We have updated the paper to include these details.

---

> > ### Comment · Reviewer_yDcV · 2025-04-05
> >
> > Thanks for the rebuttal. After reading the rebuttal and other reviews, most of my concerns are addressed. I change my Overall Recommendation to weak accept.

---

> > > ### Author Response · Authors · 2025-04-05
> > >
> > > Thank you again for your engagement and constructive feedback, which has helped us improve our paper. We sincerely appreciate your raised score.

---

### Official Review · Reviewer_nrtL · 2025-03-11

**Overall Recommendation:** 4

**Summary:**

This paper addresses cross-domain visual localization challenges by proposing a novel data generation framework based on 3D Gaussian Splatting (3DGS). The key contributions include: (1) a cross-domain 3DGS that models photometric variations and mitigates dynamic object interference, (2) a text-guided image editing model for addressing long-tail distribution problems, (3) an anchor-based method for high-quality dataset generation, and (4) a positional attention mechanism for cross-camera data ambiguities. Experiments show improvement across domains compared to baselines.

**Claims And Evidence:**

This paper clearly identify limitations of single-domain methods and provide a convincing rationale for the method. Experimental results on the 360Loc dataset with different domains demonstrate improvements.

**Essential References Not Discussed:**

The paper would benefit from discussing several relevant works in the areas of neural rerendering for unconstrained photo collections and cross-domain visual localization:

1. Neural Rerendering for Unconstrained Photo Collections:
   - Neural Rerendering in the Wild [CVPR'19]
   - NeRF in the Wild [CVPR'21]
   - Ha-NeRF [CVPR'22]
   - Neural Scene Chronology [CVPR'23]
   - NeRF On-the-go [CVPR'24]
   - SpotLessSplats [TOG'25]

2. Cross-Domain Visual Localization:
   - Adversarial training for adverse conditions: Robust metric localisation using appearance transfer [ICRA'18]
   - Night-to-Day Image Translation for Retrieval-based Localization [ICRA'19]
   - Retrieval-based localization based on domain-invariant feature learning under changing environments [IROS'19]
   - Adversarial feature disentanglement for place recognition across changing appearance [ICRA'20]
   - Place Recognition under Occlusion and Changing Appearance via Disentangled Representations [ICRA'23]

Including these references would strengthen the paper by providing a more comprehensive context for the proposed methods and highlighting the novelty of the approach in relation to existing work.

**Experimental Designs Or Analyses:**

The use of a dataset that includes multiple camera domains and varying lighting conditions and construction of a long-tailed distribution dataset effectively simulates real-world scenarios. The reported  improvement demonstrates the effectiveness.

**Methods And Evaluation Criteria:**

Using 3DGS for high-fidelity image synthesis and introducing learnable photometric embeddings are effective to handling variations. The evaluation on the 360Loc dataset with different camera types and lighting conditions provides a realistic and diverse experiment.

**Other Comments Or Suggestions:**

Figure 5 layout should be improved with a horizontal arrangement: GT, 360, fisheye3, fisheye2, fisheye1, pinhole.

**Other Strengths And Weaknesses:**

**Strengths:**
- Addresses a relevant real-world problem
- Innovative cross-domain data generation framework
- Creative use of text-guided image editing for data augmentation
- Effective positional attention mechanism for cross-camera data

**Weaknesses:**
- Limited discussion of related work

**Questions For Authors:**

No

**Relation To Broader Scientific Literature:**

Building upon advances in neural rerendering for unconstrained photo collections, this paper makes a contribution by leveraging these techniques to enhance visual localization through cross-domain image generation.

**Theoretical Claims:**

The theoretical aspects are sound.

---

> ### Author Rebuttal · Authors · 2025-03-30
>
> **1. Limited discussion of related work**
>
> Thank you for your valuable suggestion. We have expanded the related work section in the revised manuscript by incorporating the recommended references to better highlight the novelty of our proposed method. The revised portion of the related work is shown below:
>
> Recent advances in neural rendering have enabled the 3D reconstruction from unconstrained photo collections. Neural Rerendering in the Wild [1] combines traditional 3D reconstruction with neural networks to handle unconstrained scenes. Extensions to the NeRF [2,3] further address challenges in uncontrolled scenes by embedding appearance information and transient uncertainty. In addition, methods such as Ha-NeRF [4] and Neural Scene Chronology [5] focus on capturing temporal variations by employing modules for appearance hallucination and temporal step function encoding. However, their slow training and rendering make large-scale scene modeling and data generation time-consuming. Additionally, the limited parameters of NeRF hinder its ability to effectively represent large outdoor scenes. Recently, 3DGS-based methods have garnered attention due to their faster optimization and rendering efficiency compared to NeRF. Among them, SpotLessSplats [6], GS-W, WE-GS, and WildGaussians have shown potential in modeling appearance variations and dynamic objects in wild scenes. However, they cannot explicitly control photometric properties and are constrained by the performance bottlenecks of pre-trained detectors. Existing methods are also not suitable for scenes with long-tail distribution problems. In this paper, we propose a method that models appearance variations by explicitly encoding photometric histograms, mitigates the impact of dynamic objects without relying on pre-trained detectors, and employs a fine-tuned image editing model to effectively address the long-tail distribution problem.
>
> Several studies have explored cross-domain visual localization under pinhole camera settings. To enhance localization performance, [7] uses invertible generators to produce synthetic images, while [8] converts nighttime images to a more discriminative daytime representation. Other methods focus on learning domain-invariant features to bridge the gap between varying environmental conditions [9]. Additionally, several works [10,11] advocate for disentangling image representations into separate codes that isolate place-specific cues from appearance and occlusion factors, ensuring reliable place recognition. However, these methods are limited to single-camera localization and rely on image retrieval-based localization. Compared to APR methods, such approaches suffer from significantly higher computational costs and storage requirements due to the need to construct and maintain a retrieval database. This paper enhances APR-based cross-domain localization, including cross-camera scenarios, by proposing a novel cross-domain image generation method. Unlike the prior method with limited image augmentation, our method facilitates diverse cross-domain image generation.
>
> [1]Neural Rerendering in the Wild [CVPR'19]
>
> [2]NeRF in the Wild [CVPR'21]
>
> [3]Ha-NeRF [CVPR'22]
>
> [4]Neural Scene Chronology [CVPR'23]
>
> [5]NeRF On-the-go [CVPR'24]
>
> [6]SpotLessSplats [TOG'25]
>
> [7]Adversarial training for adverse conditions: Robust metric localisation using appearance transfer [ICRA'18]
>
> [8]Night-to-Day Image Translation for Retrieval-based Localization [ICRA'19]
>
> [9]Retrieval-based localization based on domain-invariant feature learning under changing environments [IROS'19]
>
> [10]Adversarial feature disentanglement for place recognition across changing appearance [ICRA'20]
>
> [11]Place Recognition under Occlusion and Changing Appearance via Disentangled Representations [ICRA'23]
>
> **2. Figure 5 layout should be improved with a horizontal arrangement: GT, 360, fisheye3, fisheye2, fisheye1, pinhole.**
>
> Thank you for your valuable suggestion. We have revised the layout of Figure 5 to present the images in a horizontal arrangement as suggested. The manuscript has been updated accordingly.

---

> > ### Comment · Reviewer_nrtL · 2025-04-04
> >
> > Thanks for the rebuttal. After reading the other reviews and the rebuttal, I recommend accepting this paper. I highly encourage the authors to revise the paper to incorporate the rebuttal, either in the main text or in the supplementary materials.

---

> > > ### Author Response · Authors · 2025-04-05
> > >
> > > Thank you very much for your positive feedback and kind recommendation. We will carefully revise the paper to incorporate the clarifications and improvements discussed in the rebuttal.

---

### Official Review · Reviewer_zXjH · 2025-03-12

**Overall Recommendation:** 3

**Summary:**

This paper proposes a novel cross-domain data generation framework to enhance visual localization in scenarios with significant appearance variations (e.g., lighting conditions, camera types). The key contributions include a cross-domain 3D Gaussian Splatting framework, a text-guided image editing model, an anchor-based dataset generation method, and a positional attention mechanism. Experiments on the 360Loc benchmark demonstrate state-of-the-art performance.

## update after rebuttal
After carefully reading the reviews from other reviewers and the authors' rebuttal, I have decided to maintain my original rating (Weak Accept).

**Claims And Evidence:**

The claims are generally well-supported.

**Essential References Not Discussed:**

None.

**Experimental Designs Or Analyses:**

1. What is the APR method used in the proposed approach—PN, MS-T, or other methods?

2. The ablations are mainly evaluated based on rendering quality. It would be beneficial to compare localization performance to demonstrate the relationship between rendering quality and localization performance.

**Methods And Evaluation Criteria:**

1. The integration of 3DGS with photometric embeddings and dynamic suppression is novel and appropriate for modeling cross-domain variations.

2. The 360Loc dataset is suitable for evaluation.

**Other Comments Or Suggestions:**

None.

**Other Strengths And Weaknesses:**

Strengths:
1. Novel framework for cross-domain data generation.
2. SOTA performance on the 360Loc benchmark.

**Questions For Authors:**

Please refer to "Experimental Designs Or Analyses".

**Relation To Broader Scientific Literature:**

This work builds on 3D Gaussian Splatting, image editing and visual localization.

**Theoretical Claims:**

N/A.

---

> ### Author Rebuttal · Authors · 2025-03-30
>
> **1. What is the APR method used in the proposed approach—PN, MS-T, or other methods?**
>
> Sorry for the unclear details.  Our proposed approach employs the MS-T method for absolute pose regression (APR).
>
> **2. The ablations are mainly evaluated based on rendering quality. It would be beneficial to compare localization performance to demonstrate the relationship between rendering quality and localization performance.**
>
> We sincerely appreciate your insightful suggestion. To evaluate the impact of rendering quality,  we conduct a quantitative experiment on the Atrium scene. Specifically, we compare the original Scaffold-GS (which produces lower-quality renderings compared to our cross-domain 3DGS) combined with our proposed data generation and positional attention mechanism.
> | Method                  | Atrium_Day | Atrium_Night | Average     |
> |-------------------------|------------|--------------|-------------|
> | Scaffold-GS             | 3.5/18.2   | 8.4/37.5     | 5.95/27.85  |
> | **Cross-domain 3DGS（Ours）** | **2.6/14.5**   | **2.2/12.8**     | **2.4/13.65**   |
>
>  As shown in the table, localization performance drops when using the lower-quality images generated by Scaffold-GS. This result demonstrates a positive correlation between rendering quality and localization accuracy.

---

### Decision · Program_Chairs · 2025-05-01

**Decision:**

Accept (poster)

**Comment:**

The ratings for this paper are one accept (Reviewer nrtL) and three weak accepts. Both Reviewer zXjH and Reviewer nrtL agree that the proposed method is novel and appreciate the good performance and the proposed novel cross-domain data generation framework. Both Reviewer nrtL and Reviewer MirD acknowledge that the paper is addressing a real-world problem. The major concerns of Reviewer yDcV are about its insufficient experiments and incremental contributions. The rebuttal satisfactorily addressed most of these concerns, and Reviewer yDcV raised the rating to weak accept. Therefore, the ACs recommend acceptance.